# Propensity Score Analysis of the Utility of Supervised Perioperative Abdominal Wall Exercises for the Prevention of Parastomal Hernia

**DOI:** 10.3390/nursrep15020062

**Published:** 2025-02-08

**Authors:** Victoria Alejandra López-Callejón, Amparo Yuste-Sanchez, Mayed Murad, Rut Navarro-Martínez, Leticia Pérez-Santiago, José Martín-Arevalo, David Moro-Valdezate, Vicente Pla-Martí, David Casado-Rodriguez, Alejandro Espí-Macías, Stephanie García-Botello

**Affiliations:** 1Colorectal Surgery Unit, Biomedical Research Institute INCLIVA, Hospital Clínico Universitario, 46010 Valencia, Spain; lopez_viccala@gva.es (V.A.L.-C.);; 2Physiotherapy Unit, Department of Rehabilitation and Physiotherapy, Hospital Clínico Universitario, 46010 Valencia, Spain; yuste_mam@gva.es (A.Y.-S.); murad_may@gva.es (M.M.); 3Care Research Group (INCLIVA) Clinic Hospital of Valencia, 46010 Valencia, Spain; 4Department of Surgery, Universidad de Valencia, 46010 Valencia, Spain

**Keywords:** abdominal exercise, propensity score matching, supervised, parastomal hernia, prophylactic measures

## Abstract

Retrospective studies have suggested that performing perioperative abdominal wall exercises may decrease the incidence of parastomal hernias. **Objectives:** This study seeks to assess the usefulness of supervised preoperative and postoperative abdominal wall exercises in the prevention of parastomal hernia. **Methods:** An observational study of patients who underwent a stoma, temporary or permanent, between January 2019 and December 2020, was performed. Minimum follow-up was 12 months. During the first 12 months of recruitment, patients were enrolled on a consecutive basis and assigned to the control group, and the remaining patients were assigned to the intervention group. A propensity score matching was performed to obtain totally comparable groups. A set of exercises was designed by the Rehabilitation Department, and their performance was supervised by physiotherapists and stoma therapists. The diagnosis of parastomal hernia was made by physical examination and computed axial tomography. Descriptive statistics of the study group were performed. Subsequently, prediction models for the occurrence of parastomal hernia were created based on binary logistic regression and classification trees. **Results:** After propensity matching and inclusion criteria, 64 patients were included (colostomy: *n* = 39, ileostomy: *n* = 25). Independent prognostic variables for parastomal hernias in colostomy were age (*p* = 0.044) and perioperative exercises (*p* = 0.003). The binary logistic regression model based on these variables gave an AUC of 97.6. The classification tree model included only perioperative exercises with an AUC of 92.5%. In the case of ileostomy, perioperative exercises were the only independent prognostic variable identified. The classification-tree-based model reported an AUC of 84%. **Conclusions**: The performance of supervised abdominal wall training and strengthening exercises may be useful in the prevention of parastomal hernias.

## 1. Introduction

Despite important advances in the management of digestive and urinary pathology, a significant percentage of patients still require a definitive stoma for the treatment of their disease. The most frequent postoperative complication of a stoma is parastomal hernia (PSH), with an unclear incidence, due to different definitions of this problem and the diagnostic method used. According to the General Nursing Council, the incidence of stoma creation in Spain is approximately 16,000 new cases annually, and the current number of ostomized patients in the country is estimated at 70,000 cases.

The frequency of PSH varies according to the type of stoma performed. The highest incidence would correspond to terminal colostomies performed in Hartmann’s procedures or abdominoperineal amputations, with rates reaching up to 40% at two years [1], followed by ileostomies or Brooke urostomies, with figures ranging from 10 to 35% [2,3,4]. Identified risk factors for developing a PSH include body mass index (BMI), female sex, preoperative albumin level, prolonged surgical times, postoperative renal failure, patients with a history of previous laparotomies, tobacco consumption, use of prophylactic mesh, type of hospital where the procedure is performed, and parastomal wound infection [2,3,5,6]. PSH prediction systems have also been developed to identify patients at higher risk [3].

Published works on preventive measures advised by stoma therapists during the perioperative period, such as physical exercise and clothing, suggest that they could decrease the risk of PSH [3,7,8,9]. These publications have shown that patients who underwent a series of abdominal wall training exercises and those who engaged in regular exercise had a lower incidence of PSH. Recent publications have suggested a relationship between possible atrophy of the abdominal wall musculature and the development of PSH [10], leading to speculation that abdominal wall muscle training may be associated with a reduced risk of PSH.

The aim of this study was to assess whether performing perioperative abdominal wall training exercises could reduce the risk of developing postoperative parastomal herniation.

## 2. Materials and Methods

### 2.1. Study Design and Population

This was a prospective observational study involving patients undergoing stoma surgery between 1 January 2019 and 31 December 2020. Patients were consecutively enrolled in the control group for the first 12 months and the exercise group for the remaining months. All patients referred for stoma marking were offered participation in the study. Therefore, only the stoma therapists and physiotherapists were aware of the group assignment. This study included patients aged 18 years and older who underwent stoma surgery due to a preoperative diagnosis of colorectal or bladder cancer, both in elective and urgent settings, with a minimum follow-up period of 12 months. Patients with psychiatric history or cognitive impairments, patients with mobility restrictions, patients with prophylactic meshes, cases resulting in death, and patients who underwent intestinal transit reconstruction before completing the follow-up were excluded.

The study was conducted in accordance with the guidelines of the Declaration of Helsinki, and the study protocol was approved by the local Ethics Committee (Hospital Clinic de Valencia, reference number: 2019/170). All participants gave written informed consent before being included in the study.

### 2.2. Sample Size

The sample size was calculated based on preliminary findings from the Coloproctology Unit. The method employed was a two-tailed comparison of proportions between the two study groups using a 1:1 ratio. The observed incidence of parastomal hernias at 12 months was 52%, and an expected reduction of 60% was anticipated with the abdominal wall strengthening exercise. With a significance level (alpha) of 5%, a power (beta) of 80%, and an estimated dropout rate of approximately 10%, the calculated sample size was 48 cases per group.

### 2.3. Data Collection

The study variables included patient-dependent factors such as age, preoperative hemoglobin, preoperative albumin, body mass index (BMI), arterial hypertension (HTN), diabetes mellitus, smoking history, physical status (regular physical exercise: at least 150 min per week), and primary diagnosis. Surgical-technique-related variables were also included, such as the surgical approach (open or laparoscopic), wound contamination grade, and postoperative hemoglobin levels. Complications were recorded according to the Clavien–Dindo classification, with specific documentation of wound infections, postoperative intraperitoneal complications (hemoperitoneum, diffuse peritonitis, intra-abdominal abscesses, paralytic ileus, intestinal ischemia, pancreatitis, cholecystitis, and urinary tract iatrogenic injuries), and stoma-related complications, including peristomal infection, necrosis, and stoma detachment. Additionally, the need for immediate postoperative reintervention and readmission was recorded. The outcome variable was the presence of parastomal hernia.

Patients were followed up on an outpatient basis at the Coloproctology and Stoma Care Unit. All patients underwent biannual CT scans following the oncology protocol for colorectal or urological cancers. In cases where clinical suspicion of parastomal hernia arose, complementary CT scans were performed to confirm the diagnosis. Both the surgeon who performed the initial operation and subsequent postoperative care, as well as the radiologist conducting the follow-up CT scans, were unaware of the patient’s group allocation.

The outcome variable was the presence of parastomal hernia. Parastomal hernia was defined as the protrusion of any intraperitoneal contents through the stoma orifice, assessed using computed tomography (CT) scanning.

The study group consisted of prospective cases in which patients received abdominal wall training through perioperative exercises, while the control group comprised retrospective cases where no additional perioperative measures were implemented. The abdominal wall training protocol was designed by the Physiotherapy and Rehabilitation Service of the hospital. Patients in the study group were provided with a set of exercises to be performed and were instructed by a physical therapist on how to perform them at home. Additionally, they were given a data collection diary in which they recorded their daily exercise sessions and the number of training sessions. A stomatherapist verified the correct performance of the exercises and their compliance.

### 2.4. Description and Pattern of Abdominal Work

A.Y.S., M.M., and R.N.M. developed an original preoperative exercise program for abdominal wall muscle strengthening with the goal of reducing the risk of PSH. The initiation of exercise was a minimum of six weeks prior to surgery, with monitoring of performance for a minimum of twelve months after surgery.

Starting from 7 basic exercises, the patient was asked to perform a minimum of 4, performing at least a series of 10 repetitions of each one. The work was performed in two sessions (morning and afternoon). If the patient performed only 4 exercises, they were not to be the same every day. The exercises were as follows:Connect with the trunk: In supine position with flexed legs and hands on the abdomen, the patient breathes very slowly, lengthening the exhalation until feeling a slight contraction of the deep abdominal musculature. This contraction should last 3 to 5 s before exhaling. This exercise works the internal, external, and transverse oblique muscles of the abdomen.Pelvic rotation: In supine position with flexed legs, the patient performs pelvic retroversion and anteversion while keeping the legs relaxed. This exercise works the anterior rectus muscle.Knee rotation: In supine position with flexed legs, the patient performs a lateral inclination of the knees with the legs bent. This exercise works internal and external obliques muscles working unilaterally.Elevation of the arm: Sitting, with the back as straight as possible, the patient slowly raises and lowers their arms in full extension. This exercise works the rectus femoris muscle.Elevation of knees: Sitting, in upright posture, the patient slowly flexes their hip with the knee flexed, initially without resistance and then applying progressive resistance by the patient with their hand. This exercise works the psoas-iliac muscle.Get up from the chair: Sitting on the edge of the chair (or bed, at first) with hands on the armrest (if available) or on the thighs, the patient performs a slight inclination of the trunk forward, looking up. Then, using this inclination as an impulse, the patient stands up, placing one foot slightly forward. This exercise works the quadriceps, gluteus maximus, and trunk erectors muscles.Foot swing: Standing with one foot slightly forward and maintaining an upright posture with a slight contraction of the abdominal area, the patient begins to rock with their feet. This exercise works balance and proprioception.

### 2.5. Statistical Analysis

A descriptive statistic of the study group was conducted. Quantitative variables were expressed as median (range) if the variable data did not follow a normal distribution and as mean ± standard deviation otherwise. The normality of the distribution of quantitative variables was assessed using graphical methods (Q-Q plot) in conjunction with the Shapiro–Wilk test. Qualitative variables were expressed as absolute frequency (relative frequency). The relationship between quantitative and qualitative variables was analyzed using non-parametric tests (Mann–Whitney U test), while the relationship between qualitative variables was assessed using the Chi-square test with calculation of odds ratios when necessary.

Propensity score matching was performed to obtain completely comparable groups with respect to the study variables. Logistic regression was used as the estimation algorithm, and the method of matching was nearest neighbor and a caliper of 0.2. Case assignment was 1:1 without restitution.

A binary logistic regression model was used to predict the occurrence of parastomal hernia. The model validation was performed through cross-validation. ROC curves and graphics of validation were calculated A model based on the CHAID classification tree with cross-validation was made. The main objective of this model was to show relations between study and outcome variables. A representation of the model and ROC curves was realized. A *p*-value ≤ 0.05 was considered statistically significant. The software used was R (version 4.3.0) for Windows 11 and IBM SPSS (version 26 to Windows 11).

## 3. Results

### 3.1. Global Analysis

During the study period, a total of 117 stomas were performed, of which 92 met the inclusion criteria for the study (Figure 1). Colostomy was performed in 53 cases (57.6%), and abdominal wall training exercises were conducted by 45 patients (48.9%). The diagnosis of PSH in the entire study group using CT was made in 50 cases (54.3%).

Patients with PSH had a higher median age (*p* = 0.001) and BMI (*p* = 0.045) compared to patients without this complication. Differences were also observed in terms of perioperative exercise performance (*p* < 0.001), the surgical technique employed (*p* = 0.007), and the type of stoma created (*p* = 0.001) (Table 1).

However, when comparing the group of patients who performed abdominal wall training exercises, statistically significant differences were observed in terms of age (*p* = 0.038), history of previous interventions (*p* = 0.047), surgical approach (*p* = 0.021), surgical technique (*p* = 0.005), and type of stoma created (*p* = 0.006) (Table 2).

After performing propensity score matching based on variables that differed between the two groups (age, surgical approach (laparoscopic), surgical technique performed, and type of stoma), a new group of 64 patients was obtained, fully comparable in terms of all study variables (Figure 1). The overall analysis of this group showed that age (*p* = 0.044), BMI (*p* = 0.014), ASA surgical risk assessment (*p* = 0.013), and perioperative exercise performance (*p* < 0.001, OR: 0.124, 95% CI: 0.049–0.316) were variables related to PSH. Independent prognostic factors for PSH were age (*p* = 0.040) and perioperative exercise performance (*p* < 0.001). The obtained binary logistic regression model showed a sensitivity of 88.3%, a specificity of 89.3%, an accuracy of 89.1%, and an AUC of 94.4 ± 0.031 (95% CI: 90.5%–1) (Figure 2a). Cross-validation of the model yielded a mean absolute error of 0.032 and a mean squared error of 0.002 (Figure 2b).

In the classification tree model of PSH and study variables (Figure 3a), perioperative exercise was the first variable introduced. In the group of patients who performed exercises, the occurrence of postoperative stoma infection was associated with a higher number of PSH cases. In the case of the group of patients who did not perform exercises, the variable of tobacco consumption was included, and within the subgroup of tobacco consumers, ASA III patients had the highest risk of PSH. This model had a sensitivity of 83.3%, a specificity of 100%, an accuracy of 90.6%, and an AUC of 96.1% ± 0.023% (95% CI: 91.6%–1) (Figure 3b).

Following this, a subanalysis of PSH risk based on the type of stoma created was performed (Table 3).

### 3.2. Colostomy Analysis

After propensity score matching, a total of thirty-nine colostomies were performed. Statistical analysis revealed that age (*p* = 0.26), BMI (*p* < 0.001) and perioperative exercises (*p* < 0.001, OR: 0.089, 95% CI: 0.024–0.339) were variables related to PSH (Table 4).

Among these variables, age (*p* = 0.044) and perioperative exercises (*p* = 0.003) were independent prognostic factors for the occurrence of PSH. The binary logistic regression model based on these variables showed a sensitivity of 95.8%, a specificity of 93.3%, an accuracy of 94.9%, and an AUC of 97.6 ± 0.023% (95% CI: 93–100) (Figure 4a). Cross-validation of the model yielded a mean absolute error of 0.016 and a mean squared error of 0.028 (Figure 4b). The classification tree model included only perioperative exercise performance. This model had a sensitivity of 91.7%, a specificity of 93.3%, an accuracy of 92.3%, and an AUC of 92.5% ± 0.5 (95% CI: 82.7–100).

### 3.3. Ileostomy Analysis

The number of ileostomies performed in the matched cases group was twenty-five. The only variable related to the occurrence of PSH was perioperative exercises (*p* = 0.001, OR: 0.197, 95% CI: 0.054–0.713), which was also an independent prognostic factor (*p* = 0.002). The classification tree model based on CHAID showed that the only included variable was perioperative exercise performance. This model had a sensitivity of 83.3%, a specificity of 84.6%, an accuracy of 84%, and an AUC of 84% ± 0.087 (95% CI: 67–100%).

## 4. Discussion

In this study, factors initially identified as related to the occurrence of PSH were age, BMI, perioperative abdominal exercise training, surgical technique, and stoma type. These results confirm previous studies that also found female gender, ASA status, the presence of aneurysms, and paralytic ileus to be associated with PSH occurrence [6].

The prevention of parastomal hernia occurrence is still an unresolved issue. Various technical options have been proposed to reduce the incidence of PSH, such as the SMART technique [11,12], extraperitoneal stomas [13,14,15,16,17], transrectal location [18], stoma size [19], or stoma incision type [20]. However, there is no consensus in the clinical guidelines regarding their utility due to a lack of significant evidence [1,21,22]. The use of prophylactic meshes for PSH prevention is highly controversial, with no clear evidence of their utility. Scientific societies support the use of prophylactic meshes to prevent PSH occurrence [1,21,22] with a moderate-to-strong level of evidence. However, recent studies and clinical trials have shown no significant differences in PSH incidence with the use of prophylactic meshes, raising questions about their routine use [23,24,25,26,27,28].

The abdominal wall musculature has likely been underestimated in ostomy patient management guidelines [1,21,22,29]. Atrophy of the abdominal wall and elevated BMI have been identified as risk factors for PSH development [10]. Consequently, it is logical to assume that strengthening this musculature should decrease the risk of PSH [30]. Previous nursing studies have shown that postoperative exercise training is associated with a lower incidence of PSH. However, these studies lacked supervision or systematic exercise programs [7,9,30,31,32]. In this study, both pre- and postoperative abdominal wall exercise training was designed by the Rehabilitation Service, and exercise performance was supervised by nurses from the Stoma Therapy Unit. Additionally, the occurrence of PSH was assessed by an expert radiologist to avoid bias or interpretation errors, rather than relying on patient or stoma therapist self-reports, as in previous studies.

The incidence of PSH in this study was 54.3%, like in reports by other authors [33,34,35]. Previous studies have shown that CT scans diagnose more cases of PSH compared to physical examination, with approximately 64% of cases coinciding [34,36]. In this study, the incidence of PSH was higher in the colostomy group compared to the ileostomy group (61.54% vs. 48%), consistent with previous findings [37]. As a result, colostomies and ileostomies (digestive or urologic) were studied independently. It was not possible to study digestive ileostomies and urostomies separately due to the limited sample size. Previous studies reported a 30% incidence of PSH in urostomies with 12 months of follow-up [38]. It is possible that this incidence would have been higher with a longer follow-up period.

For colostomies, age, BMI, and perioperative abdominal exercises were identified as related factors, with only age and perioperative abdominal exercises being independent prognostic factors. The classification tree showed that patients with a lower likelihood of developing PSH were those who performed exercises and did not have infectious stoma complications. Interestingly, the highest-risk group consisted of patients without a history of tobacco use and ASA III. In the case of ileostomies, perioperative abdominal wall training was the only identified risk factor. Some authors have reported that in the case of urostomies, factors associated with PSH were female gender, diabetes, COPD, and elevated BMI [38].

In this study, it was observed that abdominal wall training exercises were associated with a lower incidence of PSH and were an independent prognostic factor for this complication regardless of the type of stoma performed. These results support previous studies that indicated the potential utility of abdominal wall training exercises in preventing the occurrence of PSH [7,9,30,31,32]. Additionally, it has also been noted that patients who engage in regular physical exercise have a lower incidence of PSH compared to those with a more sedentary lifestyle (30). The incidence of PSH in the group of patients who performed abdominal wall exercises falls within the range of clinical trials (0–52%) included in recent meta-analyses [39,40,41,42,43,44,45,46,47,48].

The originality of this study lies in the supervised design and implementation of a series of abdominal wall training exercises aimed at reducing the risk of PSHs. The use of propensity score matching techniques helped reduce biases and subsequently allowed for a detailed study of patients in subgroups according to the type of stoma performed.

PSM reduces confounding in observational studies, but it has important limitations. It cannot account for unmeasured confounders, is dependent on the correct specification of the propensity score model, and often results in a loss of sample size, which reduces the power of the study. Poor overlap in propensity scores can limit matching and introduce bias. Balancing on the score itself may not ensure covariate balance. PSM also has problems with time-dependent confounding and post-treatment variables.

However, the limitations of this study include its observational nature, loss of cases in follow-up that reduced the sample size, the limited number of patients with ileostomy, and the limited follow-up of 12 months, although this follow-up period is the most reported in clinical trials and meta-analyses.

The results of this study suggest the possibility of conducting new prospective, randomized, multicenter studies to confirm these findings, and if confirmed, it could be a complementary measure to help select patients at higher risk who would benefit most from the use of a prophylactic mesh to reduce the risk of PSH, within prehabilitation and ERAS programs.

## 5. Conclusions

The performance of supervised perioperative abdominal wall training and strengthening exercises may be useful in the prevention of parastomal hernias. The findings of this study have significant implications for nursing practice, particularly for nurses from the Stoma Therapy Unit, who play a vital role in the postoperative care and rehabilitation of stoma patients. By demonstrating that supervised abdominal wall training exercises during the perioperative period can significantly reduce the incidence of PSH, stoma therapists can incorporate these exercises into patient care protocols. This proactive approach not only improves outcomes for patients but also empowers them through education and active participation in their recovery process.

In addition, these findings highlight the importance of interdisciplinary collaboration between nursing, physical therapists, and surgical teams, fostering comprehensive care that addresses both rehabilitation and the psychosocial support needed for stoma patients. Together, these strategies contribute to optimizing patient care and implementing effective preventive measures against complications associated with stomas.

## Figures and Tables

**Figure 1 nursrep-15-00062-f001:**
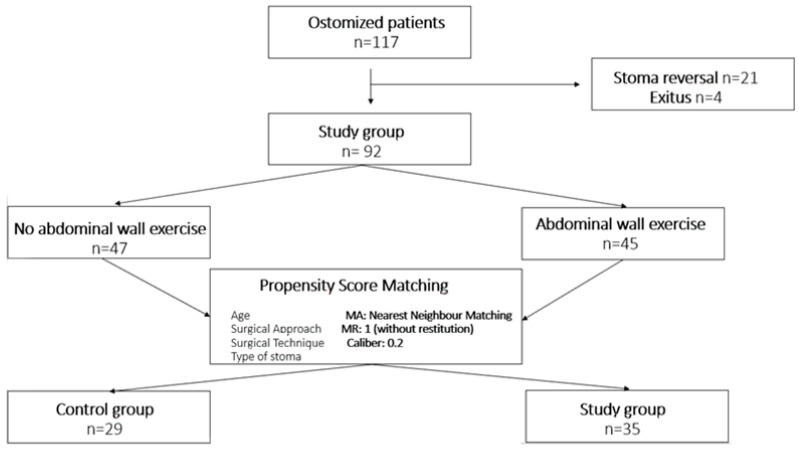
Flowchart of study patients with propensity score matching.

**Figure 2 nursrep-15-00062-f002:**
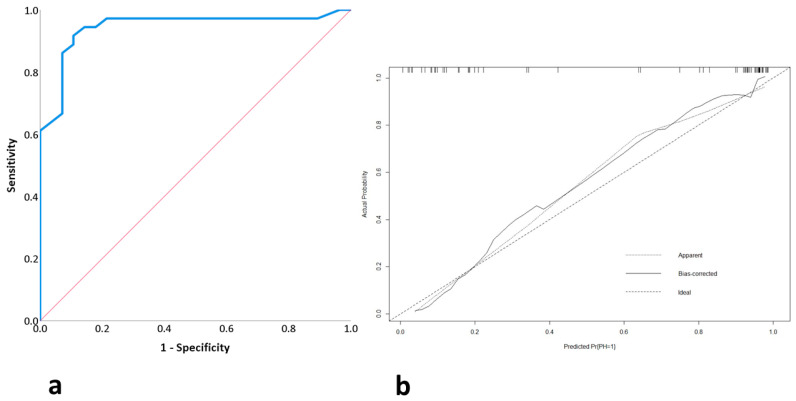
(**a**) ROC curve of logistic binary model of study group after matching cases. (**b**) Cross-validation of same model.

**Figure 3 nursrep-15-00062-f003:**
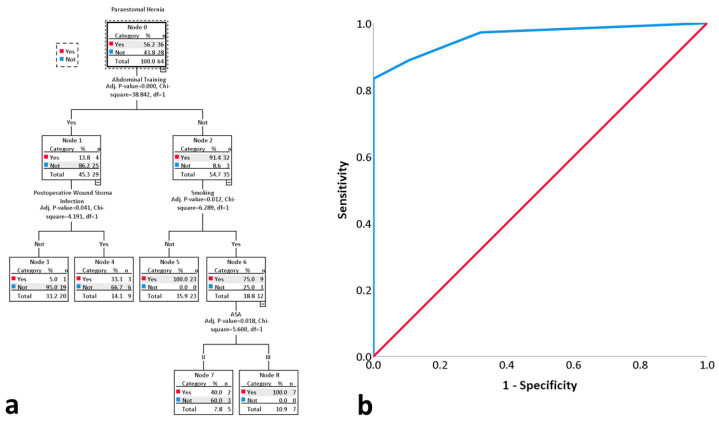
(**a**) ROC curve of CHAID classification tree model of study group after matching cases. (**b**) ROC curve of same model.

**Figure 4 nursrep-15-00062-f004:**
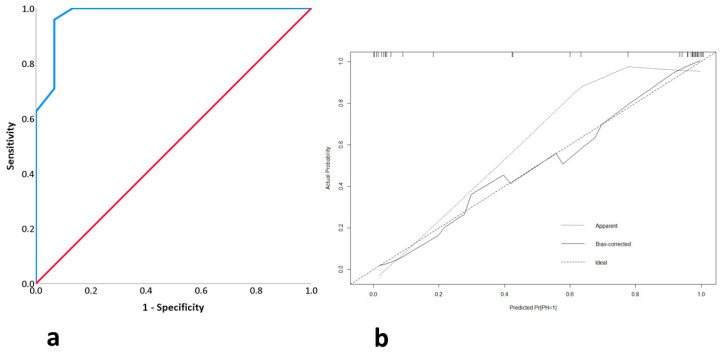
(**a**) ROC curve of logistic binary model of colostomy group after matching cases. (**b**) Cross-validation of same model.

**Table 1 nursrep-15-00062-t001:** Epidemiological and clinical characteristics of the study group before propensity score matching.

	Parastomal Hernia 54.35% (*n* = 50)	Non-Hernia 45.65% (*n* = 42)	*p*-Value
Age (years)	76 (47)	66 (59)	**0.001**
IMC (kg/m^2^)	26.39 (29.61)	25.1 (15.9)	**0.045**
Hemoglobin (mg/dL)	13.5 (8.4)	13.45 (6.5)	0.440
Albumin (g/dL)	4.3 (2.1)	4.1 (2.8)	0.189
Diabetes	16 (32%)	7 (16.7%)	0.146
Smoking	16 (32%)	14 (33.3%)	1
Hypertension	16 (32%)	15 (35.7)	0.825
Previous surgery	16 (32%)	9 (21.4%)	0.347
Physical status active	18 (36%)	23 (54.8%)	0.093
ASA			0.323
I	-	1 (2.4%)	
II	28 (56%)	29 (69%)	
III	19 (38%)	11 (26.2%)	
IV	3 (6%)	1 (2.4%)	
Abdominal training	6 (12%)	39 (92.9%)	**<0.001**
Elective surgery	44 (88%)	34 (81%)	0.393
Surgical approach (laparoscopic)	31 (62%)	20 (47.6%)	0.208
Operative time (minutes)			0.875
≤120 min	8 (16%)	5 (11.9%)	
121–240 min	16 (32%)	14 (33.3%)	
241–360 min	18 (36%)	14 (33.3%)	
>360 min	8 (16%)	9 (21.4%)	
Diagnosis			0.355
Malignant neoplasm	42 (84%)	29 (69%)	
IBD	4 (8%)	4 (9.5%)	
Peritonitis	-	2 (4.8%)	
Entero-genital or entero-urinary fistula	2 (4%)	2 (4.8%)	
Benign neoplasm	1 (2%)	4 (9.5%)	
Ischemic colitis	1 (2%)	1 (2.4%)	
Surgical technique			**0.007**
Abdominoperineal resection	23 (46%)	7 (16.7%)	
Anterior rectum resection	8 (16%)	13 (31%)	
Radical cystectomy	2 (4%)	7 (16.7%)	
Subtotal colectomy with ileostomy and mucosal fistula	1 (2%)	3 (7.1%)	
Total colectomy with Brooke ileostomy	2 (4%)	2 (4.8%)	
Hartmann’s procedure	13 (26%)	5 (11.9%)	
Colostomy and mucosal fistula	1 (2%)	4 (9.5%)	
Colectomy with double colostomy	-	1 (2.4%)	
Colostomy	37 (74%)	17 (40.5%)	**0.001**
Postoperative complications			
Surgical wound infection	7 (14%)	6 (14.3%)	1
Paralyticus ileus	16 (32%)	13 (31%)	1
Parastomal wound infection	12 (24%)	10 (23.8%)	1

Data are expressed mean (SD) and *n* (%) of each subgroup. Significant results are indicated in bold (*p* < 0.05).

**Table 2 nursrep-15-00062-t002:** Comparison of epidemiological and clinical characteristics of the study group according to the performance of abdominal wall muscle training exercises before and after matching cases.

	Pre-Propensity Score Matching			Propensity Score Matching		
	Abdominal Training *n* = 45 (48.91%)	Non-Exercise *n* = 47 (51.09%)	*p*-Value	Abdominal Training *n* = 29 (45.3%)	Non-Exercise *n* = 35 (54.7%)	*p*-Value
Age (years)	68 (66)	75 (44)	**0.038**	68 (45)	68 (44)	0.491
IMC (kg/m^2^)	25.4 (15.9)	26.35 (29.61)	0.43	25.1 (15.9)	26.35 (29.61)	0.161
Hemoglobin (mg/dL)	13.4 (6.7)	13.5 (8)	0.154	13.5 (5.9)	13.6 (7.6	0.240
Albumin (g/dL)	4.1 (2.8)	4.3 (2.1)	0.465	4.1 (2.6)	4.3 (2.1)	0.622
Diabetes	11 (24.4%)	12 (25.5%)	1	5 (17.2%)	7 (20%)	1
Smoking	14 (31.1%)	16 (34%)	0.826	10 (24.5%)	12 (34.3%)	1
Hypertension	16 (35.6%)	15 (31.9%)	0.826	11 837.9%)	10 (28.6%)	0.593
Previous surgery	8 (17.8%)	17 (36.2%)	0.062	5 (17.2%)	15 (42.9%)	**0.033**
Physical status Active	23 (51.1%)	18 (38.3%)	0.294	14 (48.3%)	15 (42.9%)	0.802
ASA			0.550			0.100
I	1 (2.2%)	-		1 (3.4%)	-	
II	29 (64.4%)	28 (59.6%)		23 879.3%)	20 (57.1%)	
III	14 (31.1%)	16 (34%)		5 (17.2%)	13 (37.1%)	
IV	1 (2.2%)	3 (6.4%)		-	2 (5.7%)	
Elective surgery	36 (80%)	42 (89%)	0.254	22 (75.9%)	30 (85.7%)	0.247
Surgical approach (laparoscopic)	19 (42.2%)	32 (68.1%)	**0.021**	14 (48.3%)	20 (57.1%)	0.616
Operative time (minutes)			0.717			0.632
≤120 min	7 (15.6%)	6 (12.8%)		3 (10.3%)	6 (17.1%)	
121–240 min	16 (35.6%)	14 (29.8%)		12 (41.4%)	10 (28.6%)	
241–360 min	13 (28.9%)	19 (40.4%)		9 (31%)	14 (40%)	
>360 min	9 (20%)	8 (17%)		5 (17.2%)	5 (14.3%)	
Diagnosis			0.369			0.586
Malignant neoplasm	31 (68.9%)	40 (85.1%)		21 (72.4%)	29 (82.9%)	
IBD	5 (11.1%)	3 (6.4%)		2 (6.9%)	2 (5.7%)	
Peritonitis	2 (4.4%)	-		1 (3.4%)	-	
Entero-genital or entero-urinary fistula	2 (4.4%)	2 (4.3%)		2 (6.9%)	2 (5.7%)	
Benign neoplasm	4 (8.9%)	1 (2.1%)		3 (10.3%)	1 (2.9%)	
Ischemic colitis	1 (2.2%)	1(2.1%)		-	1 (2.9%)	
Surgical technique			**0.005**			0.424
Abdominoperineal resection	7 (15.6%)	23 (48.9%)		6 (20.7%)	14 (40%)	
Anterior rectum resection	13 (28.9%)	8 (17%)		7 (24.1%)	8 (22.9%)	
Radical cystectomy	7 (15.6%)	2 (4.3%)		4 (13.8%)	2 (5.7%)	
Subtotal colectomy with ileostomy and mucosal fistula	4 (8.9%)	-		1 (3.4%)	-	
Total colectomy with Brooke ileostomy	2 (4.4%)	2 (4.3%)		1 (3.4%)	2 (5.7%)	
Hartmann’s procedure	7 (15.6%)	11 (23.4%)		6 20.7%)	8 (22.9%)	
Colostomy and mucosal fistula	4 (8.9%)	1 (2.1%)		3 (10.3%)	1 (2.9%)	
Colectomy with double colostomy	1 (2.2%)	-		1 (3.4%)	-	
Type of stoma (colostomy)	19 (42.2%)	35 (74.5%)	**0.003**	16 (56.2%)	23 (65.7%)	0.447
Parastomal hernia	6 (12%)	39 (92.9%)	**<0.001**	4 (13.8%)	32 (91.4%)	**<0.001**
	Postoperative Complications					
Surgical wound infection	6 (13.9%)	7 (14.9%)	1	4 (13.8%)	7 (20%)	0.741
Paralyticus ileus	16 (35.6%)	13 (27.7%)	0.503	10 (34.5%)	10 828.6%)	0.787
Parastomal wound infection	13 (28.3%)	9 (19.1%)	0.332	9 (31%)	6 (17.1%)	0.242

Data are expressed mean (SD) and *n* (%) of each subgroup. Significant results are indicated in bold (*p* < 0.05).

**Table 3 nursrep-15-00062-t003:** Comparison of epidemiologic and clinical characteristics of the groups according to exercise performance after empirical studies by matching cases.

	Colostomy			Ileostomy		
	Abdominal Training *n* = 16 (41%)	Non-Exercise *n* = 23 (59%)	*p*-Value	Abdominal Training *n* = 13 (52%)	Non-Exercise *n* = 12 (48%)	*p*-Value
Age (years)	66.5 (45)	68 (38)	0.301	72 (27)	70.5 (37)	0.894
IMC (kg/m^2^)	22.5 (12.7)	26.4 (27.66)	**0.001**	27.3 (14.8)	26 (11.55)	0.098
Hemoglobin (mg/dL)	13.1 (5.6)	13.6 (7.5)	0.507	13.5 (4.1)	13.75 (6.8)	0.347
Albumin (g/dL)	3.95 (2.6)	4.3 (2)	0.507	4.1 (2.4)	4.35 (1.9)	0.979
Diabetes	2 (12.5%)	4 (17.4%)	1	3 (23.1%)	3 (25%)	1
Smoking	4 (25%)	6 (26.1%)	1	6 (46.2%)	6 (50%)	1
Hypertension	4 (25%)	8 (34.8%)	0.726	7 (53.8%)	2 (16.7%)	0.097
Previous surgery	5 (31.3%)	9 (39.1%)	0.740	-	6 (50%)	**0.005**
Physical status active	8 (50%)	9 (39.1%)	0.531	6 (46.2%)	6 (50%)	1
ASA			0.242			0.367
I	-	-		1 (7.7%)	-	
II	13 (81.3%)	13 (56.5%)		10 (76.9%)	7 (58.3%)	
III	3 (18.8%)	9 (29.1%)		2 (15.4%)	4 (33.3%)	
IV	-	1 (4.3%)		-	1 (8.3%)	
Elective surgery	12 (75%)	21 (91.3%)	0.205	10 (76.9%)	9 (75%)	1
Surgical approach	7 (43.8%)	12 (52.2%)	0.748	7 (53.8%)	8 (66.7%)	0.688
Operative time (minutes)			0.649			0.084
≤120 min	3 (18.8%)	3 (13%)		-	3 (25%)	
121–240 min	7 (43.8%)	9 (39.1%)		5 (38.5%)	1(8.3%)	
241–360 min	6 (37.5%)	9 (39.1%)		3 (23.1%)	5 (41.7%)	
>360 min	-	2 (8.7%)		5 (38.5%)	3 (25%)	
Diagnosis			0.418			0.367
Malignant neoplasm	9 (56.3%)	18 (78.3%)		12 (92.3%)	11 (91.7%)	
IBD	2 (12.5%)	2 (8.7%)		-	-	
Peritonitis	-	-		1 (7.7%)	-	
Entero-genital or entero-urinary fistula	2 (12.5%)	2 (8.7%)		-	-	
Benign neoplasm	3 (18.8%)	1 (4.3%)		-	-	
Ischemic colitis	-	-		-	1 (8.3%)	
Surgical technique			0.224			0.566
Abdominoperineal resection	6 (37.5%)	14 (60.9%)		-	-	
Anterior rectum resection	-	-		7 (53.8%)	8 (66.7%)	
Radical cystectomy	-	-		4 (30.8%)	2 (16.7%)	
Subtotal colectomy with ileostomy and mucosal fistula	-	-		1 (7.7%)	-	
Total colectomy with Brooke ileostomy	-	-		1 (7.7%)	2 (16.7%)	
Hartmann’s procedure	6 (37.5%)	8 (34.8%)		-	-	
Colostomy and mucosal fistula	3 (18.8%)	1 (4.3%)		-	-	
Colectomy with double colostomy	1 (6.3%)	-		-	-	
Parastomal hernia	2 (12.5%)	22 (95.7%)	**<0.001**	2 (15.4%)	10 (83.3%)	**0.001**
	Postoperative Complications					
Surgical wound infection	2 (12.5%)	3 (13%)	1	2 (15.4%)	4 (33.3%)	0.378
Paralyticus ileus	4 (25%)	5 (21.7%)	1	6 (46.2%)	5 (41.7%)	1
Parastomal wound infection	4 (25%)	2 (8.7%)	0.205	5 (38.5%)	4 (33.3%)	1

Data are expressed mean (SD) and *n* (%) of each subgroup. Significant results are indicated in bold (*p* < 0.05).

**Table 4 nursrep-15-00062-t004:** Comparison of the epidemiological, clinical characteristics, and clinical outcomes of the groups according to the type of stoma performed and (colostomy or ileostomy) after matching cases.

	Colostomy			Ileostomy		
	Parastomal Hernia *n* = 24 (61.5%)	Non-Hernia *n* = 15 (38.5%)	*p*-Value	Parastomal Hernia *n* = 12 (48%)	Non-Hernia *n* = 13 (52%)	*p*-Value
Age (years)	72 (38)	66 (41)	**0.026**	73.5 (37)	69 (32)	0.538
IMC (kg/m^2^)	26.54 (27.66)	21.3 (12.7)	**<0.001**	26.21 (16.25)	27.1 (15.2)	0.574
Hemoglobin (mg/dL)	13.55 (7.5)	13.5 (5.6)	0.831	13.5 (6.2)	13.4 (4.1)	0.225
Albumin (g/dL)	4.3 (2)	3.8 (2.6)	0.432	4.35 (2)	4.1 (2.4)	0.406
Diabetes	5	1	0.376	4	2	0.378
Smoking	5	5	0.463	6	6	1
Hypertension	8	4	0.734	3	6	0.411
Previous surgery	9	5	1	5	1	0.073
Physical status active	9	8	0.508	5	7	0.695
Abdominal Training	2	14	**<0.001**	2	11	**0.001**
ASA			0.106			0.107
I	-	-		-	1	
II	13	13		6	11	
III	10	2		5	1	
IV	1	-		1	-	
Elective surgery	21	12	0.658	9	10	1
Surgical approach	12	7	1	6	9	0.428
Operative time (minutes)			0.626			0.084
≤120 min	4	2		3	-	
121–240 min	10	6		1	5	
241–360 min	8	7		5	3	
>360 min	2	-		3	5	
Diagnosis			0.309			0.367
Malignant neoplasm	19	8		11	12	
IBD	2	2		-	-	
Peritonitis	-	-		-	1	
Entero-genital or entero-urinary fistula	2	2		-	-	
Benign neoplasm	1	3		-	-	
Ischemic colitis	-	-		1		
Surgical technique			0.212			0.566
Abdominoperineal resection	14	6		-	-	
Anterior rectum resection	-	-		8	7	
Radical cystectomy	-	-		2	4	
Subtotal colectomy with ileostomy and mucosal fistula	-	-		-	1	
Total colectomy with Brooke ileostomy	-	-		2	1	
Hartmann’s procedure	9	5		-	-	
Colostomy and mucosal fistula	1	3		-	-	
Colectomy with double colostomy	-	1		-	-	
	Postoperative Complications					
Surgical wound infection	3	2	1	3	3	1
Paralytic ileus	7	2	0.437	5	6	1
Parastomal wound infection	4	2	1	4	5	1

Data are expressed mean (SD) and *n* (%) of each subgroup.

## Data Availability

Detailed data are available upon reasonable request to the corresponding author.

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
