# Peer review of "Propensity Score Analysis of the Utility of Supervised Perioperative Abdominal Wall Exercises for the Prevention of Parastomal Hernia"

_nursrep, 2025, doi:10.3390/nursrep15020062_

Round 1

Reviewer 1 Report

Comments and Suggestions for Authors

An interesting paper about physical activity and PSH. There are however some concern regarding your paper. 

Abstract: There you state that you divid the patients in two groups and than you say that one group was retrospective and onte prospective. This is not possible. Please rewrite.

Material - population. It is not possible to know what the population consisted of. Here again prospective study. How did you chose if a patient should be offered exercise or not? How could radiologists and surgeons be unaware of the patient group allocation?

Text in row 107-112 is also in row 116-121.

Result. How were the groups diveded? The comparision between the groups is dependent of the inclusion in the different allocation groups. Even if you have used prepensity score to be able to analyse small groups the way to decide in which group a certain patient should belong to is important. 

In the material section CT scan should be performed if PSH was suspected. Here it says that it was performed because of the primary diagnose, with a CT scan every year after surgery. 

Surgical approach, what is that? Laparoscopic/Robotic compared with open or what. A very important knowledge.

Why hemoglobin content? Not a riskfactor for PSH.

Table two what is physical status and how is it meassured? Active 14848.3%????

Propensity score "all variable that differed" Meaning what, which were...

93% PSH in the non exercise group does not make sense after one year.

Fig 3A not possible to read. 

Table 4 speaking of per centage when the groups are of 12-15 patients not suitable.

Discussion. Row 296-297 no tobacco use and smoking risk factors??

Things to highligt more

Propensity score may alsom increase bias, something about this in the discussion.

Extreme PSH values after one year. Nowadays usually around 30% after one year to increase with time.

No information about physical status before surgery and how that could be connected to the result.

Reviewer 2 Report

Comments and Suggestions for Authors

Dear authors

it is a great pleasure to be able to review such a well-thought-out and correctly conducted study, I have no major comments, congratulations on the idea and work that will probably translate into clinical practice

Verse 18-34

Please check the anatomical nomenclature, incorrect formulas "colos" - colostomy and other expressions regarding the intestinal stoma

Verse 124-128

Please check the spelling, please indicate by name who created the exercise protocol, whether it is original, has been subject to a previous taxonomy or is being used for the first time

Verse 131-150

Please consider presenting the minutes in a graphical form, this form may be more transparent

Reviewer 3 Report

Comments and Suggestions for Authors

Dear authors,

Thank you for the opportunity to review your manuscript. I have to confess that it was a real pleasure to read your paper. Congrats on your work. I have no comments and nothing to add. Please check the abstract lines 19 and 20, where there is a small inadvertence: instead of parastomal hernia, it is parasternal hernia. 

Round 2

Reviewer 1 Report

Comments and Suggestions for Authors

I think that the authors have done a good job with the manuscript. I just have a few minor hang-ups. 

material and methods; row 77. If you have one year with control material and than one year with training than of course everone involved no if the patient had trained or not.

row 99 how was regular physical exercise meassured - accelerometer, questionnaire or yes/no?

row 134. for how many weeks did the training last

fig 3, if it´s not possible to read, should you really have it there at all or present in antother way?
